# Proteomic and transcriptomic profiling reveal different aspects of aging in the kidney

Yuka Takemon[1], Joel M Chick[2,3], Isabela Gerdes Gyuricza[1], Daniel A Skelly[1], Olivier Devuyst[4], Steven P Gygi[2], Gary A Churchill[1], Ron Korstanje[1]*

[1]The Jackson Laboratory, Bar Harbor, United States; [2]Harvard Medical School, Boston, United States; [3]VividionTherapeutics, San Diego, United States; [4]Institute of Physiology, University of Zurich, Zurich, Switzerland

**Abstract** Little is known about the molecular changes that take place in the kidney during the aging process. In order to better understand these changes, we measured mRNA and protein levels in genetically diverse mice at different ages. We observed distinctive change in mRNA and protein levels as a function of age. Changes in both mRNA and protein are associated with increased immune infiltration and decreases in mitochondrial function. Proteins show a greater extent of change and reveal changes in a wide array of biological processes including unique, organ-specific features of aging in kidney. Most importantly, we observed functionally important age-related changes in protein that occur in the absence of corresponding changes in mRNA. Our findings suggest that mRNA profiling alone provides an incomplete picture of molecular aging in the kidney and that examination of changes in proteins is essential to understand aging processes that are not transcriptionally regulated.

*For correspondence:
Ron.Korstanje@jax.org

Competing interests: The authors declare that no competing interests exist.

## Introduction

Aging is characterized by a decline in physiologic function of all organs and increasing rates of disease and mortality. Kidney function is affected early in the aging process and declines progressively with age. Decreased renal volume and the loss of functioning filtration units lead to a decline in glomerular filtration rate of 5–10% per decade after the age of 35 in humans (*Glassock and Rule, 2012*). It is relatively easy and non-invasive to measure functional changes of the kidney in blood and urine (*Lindeman et al., 1985*). Moreover, kidney function significantly impacts age-related diseases in other organs including cognitive impairment (*Elias et al., 2009*). These factors make the kidney an excellent model to study organ-specific aging.

While physiological changes in kidney function are well documented, relatively little is known about the underlying molecular processes that drive age-related loss of function. Molecular profiling of normal aging kidneys at different life stages is most readily carried out in a model organism. Previous molecular profiling studies of the aging kidney have been limited to microarray based gene expression assays in young versus old kidneys of mice and rats (*Jonker et al., 2013*; *Park et al., 2016*). More recently, single cell transcriptional profiling of aging C57BL/6NJ mice has been reported for multiple tissues, including kidney (*Almanzar et al., 2020*). When aging studies are carried out in single inbred strains, they may display specific and idiosyncratic patterns of aging and thus lack generalizability beyond the particular strain studied (*Voelkl et al., 2020*). In order to capture the full range of pathologies associated with the aging kidney, a genetically diverse set of animals is required. In addition, previous studies have not attempted to relate transcriptional variation to changes in protein levels that are arguably more relevant to physiological aging.

Here we examine physiological kidney function together with mRNA and protein levels in male and female diversity outbred (DO) mice at ages 6, 12, and 18 months. The DO mice are a genetically diverse, outbred population derived from eight founder strains that together contribute >50 million well-characterized variants (*Svenson et al., 2012*). Importantly, each DO animal is genetically unique and has the potential to reveal distinct age-related pathologies. This allows us to examine the common features of age-related change that are not specific to a single genetic background. Genetic variation will drive variation in mRNA and protein abundances for the majority of genes expressed in the kidney. This provides a unique opportunity to characterize the relationship between mRNA expression and protein levels and to examine how this relationship changes with age.

## Results

### Experimental design

We carried out a cross-sectional aging study of ~600 DO mice including equal numbers of male and female mice that were aged to 6, 12, and 18 months. To evaluate kidney function, we successfully collected spot urine from 490 mice (141 at 6 months, 199 at 12 months, and 150 at 18 months) and measured albumin, creatinine, and phosphate levels. We obtained flash frozen tissue from the right kidney of 188 randomly selected mice (30 males and 33 females at 6 months; 31 males and 31 females at 12 months; and 34 males and 29 females at 18 months). We quantified mRNA by RNA-Seq and detected expression of 22,259 genes. We obtained untargeted proteomics data on the same set of 188 kidney samples and quantified 6580 proteins representing 6515 unique genes. Both mRNA and protein data were available for 6449 genes.

### Sources of variation in mRNA and protein expression

In order to characterize the main sources of variation in mRNA and protein in aging mouse kidney, we computed principle components (PCs) on the common set of 6449 genes. Data were transformed to rank-normal scores to reduce the influence of outliers.

The top four PCs for RNA explain 46.5% of variance (*Figure 1A*). The first PC explains 22.1% of variance and is strongly correlated with sex (p-value $<2.2 \times 10^{-16}$) and only weakly correlated with age (p-value = 0.050). The second PC (PC2) explains 15.3% of total variance and is not associated with sex (p-value = 0.30) or with age (p-value = 0.87). PC2 is explained by a batch effect in the RNA sequencing that we account for in subsequent analyses (see Materials and methods). The effects of age on mRNA are apparent in the third and fourth PCs but these explain 5% or less of total variation. Overall sex is a dominant factor in determining mRNA variation with minor contribution attributable to age.

The top four PCs for protein explain 39.1% of total variation (*Figure 1B*). The first PC explains 16.6% and is strongly associated with age (p-value = $5.3 \times 10^{-14}$) but not with sex (p-value = 0.24). The second PC explains 12.0% of variation. It is associated with sex (p-value < $2.2 \times 10^{-16}$) and only marginally so with age (p-value = 0.014). Effects of age and sex are apparent in the third and fourth PCs which each explain less than 5% of total variation. In contrast to mRNA, age is a dominant factor in determining protein variation with lesser but still substantial influence due to sex. We also observed that the age-specific variability of the protein PCs (especially PC1 and PC4) is greater in the 18-month animals indicating that there is greater between-animal variation in protein at later ages. In contrast, variation of mRNA is constant across age groups.

### Age-related changes in mRNA expression

To identify transcripts that change with age, we applied differential expression testing with DESeq2 (*Love et al., 2014*) (see Materials and methods) (*Supplementary file 1*). We identified 449 transcripts that showed a trend with age (adjusted p-value < 0.05) of which 426 show increasing expression with age and only 23 are decreasing. It is striking that 95% of the changes are increasing, but this is a stringent family-wise criterion that selects only transcripts with the biggest age-related changes. At a relaxed stringency using a false discovery rate criterion (FDR < 0.1), we identified 4039 transcripts that change with age and 2649 (65%) are increasing. The 426 mRNA species with the largest increase with age, we find 83 immunoglobulin genes (*Igh* and *Igk* classes). The 23

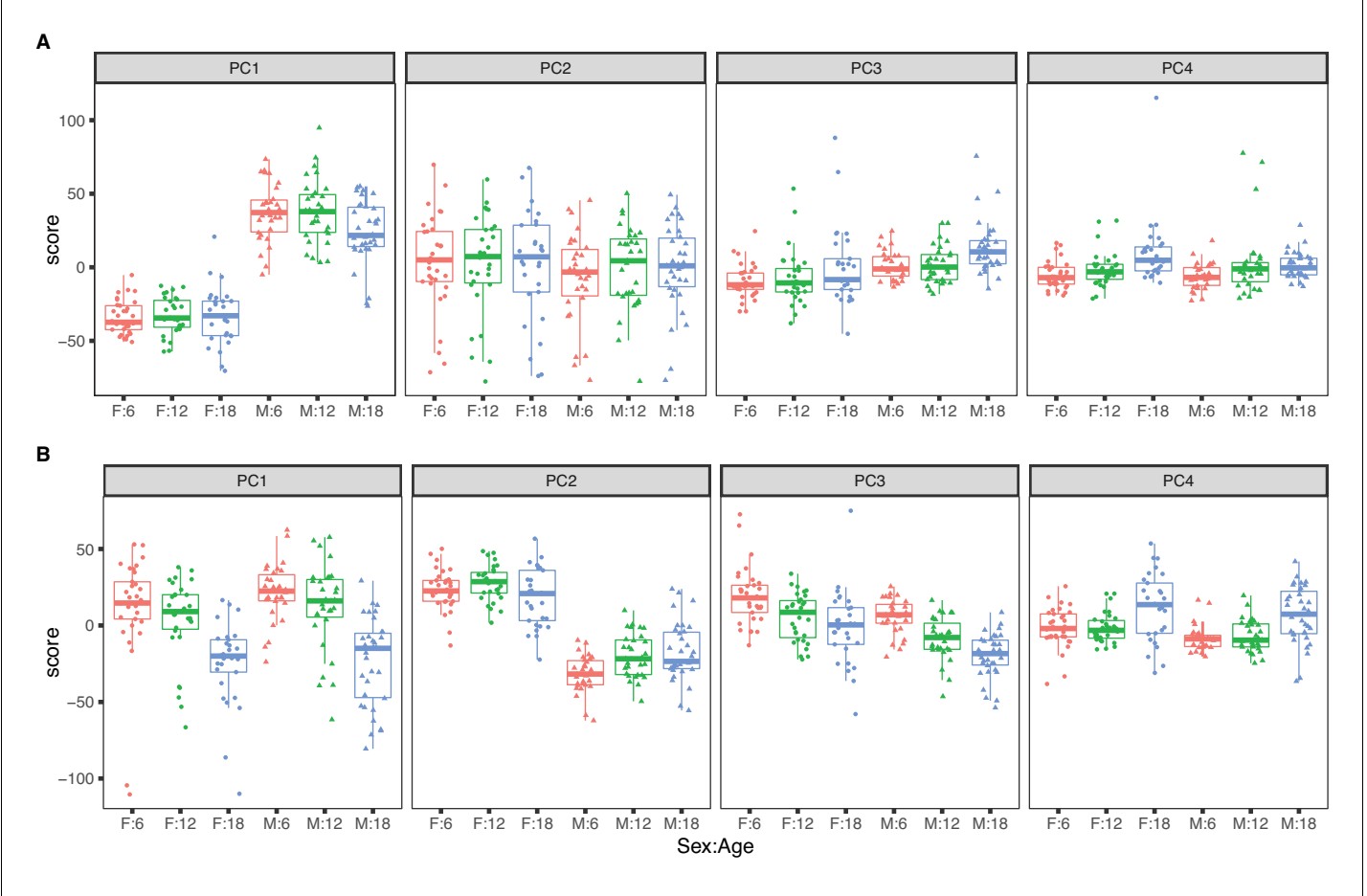

**Figure 1.** Principal component analysis for females (dots) and males (triangles) at 6 (red), 12 (green), and 18 months (blue) of age. The top four principal components for (**A**) RNA expression, with sex as the dominant factor and (**B**) protein expression, with age as the dominant factor.

decreasing mRNA species include several that encode heat shock proteins (*Hsp90aa1*, *Hsp90ab1*, *Hsph1*, and *Hspa4l*).

We evaluated the list for functional enrichment of GO categories using ClusterProfiler (*Yu et al., 2012*; *Figure 2*). We observed significant overrepresentation of genes associated with adaptive immune response (adjusted p-value = $1.3 \times 10^{-23}$), leukocyte activation (adjusted p-value = $4.9 \times 10^{-25}$), and leukocyte cell–cell adhesion (adjusted p-value = $1.4 \times 10^{-16}$). All of the top 100 enrichment categories were related to immune cell functions. Increased expression of immune and inflammatory response genes in the aging kidney has been shown in previous mRNA expression studies and is likely the result of immune cell infiltration into the kidney (*Rodwell et al., 2004*; *Melk et al., 2005a*; *Park et al., 2016*).

In order to identify the cell types involved in age-related changes, we performed in silico cell type deconvolution of our bulk RNA-Seq data using published transcriptional profiles from single cell RNA-Seq data (*Park et al., 2018*). We found a significant increase in B cell proportion (adjusted p = $6.75 \times 10^{-6}$) as well as macrophage proportion (adjusted p-value = 0.0155) with age in both sexes (*Figure 2—figure supplement 1*).

Among the mRNA with most significant age-related changes, we noted increasing expression of *Cdkn2a* (adjusted p-value = $6.1 \times 10^{-9}$) (*Figure 2—figure supplement 2*). One of the two proteins encoded by *Cdkn2a*, p16 (or p16INK4a) is a hallmark for senescence (*Hernandez-Segura et al., 2018*). However, due to low expression levels we were not able to quantify isoform-specific expression. In an in vivo study, *Cdkn2a* was found to be highly expressed in the renal cortex and associated with severity of age-associated glomerulosclerosis, tubular damage, and interstitial fibrosis (*Melk et al., 2005b*; *Bolignano et al., 2014*). We also observed increased expression of cyclin

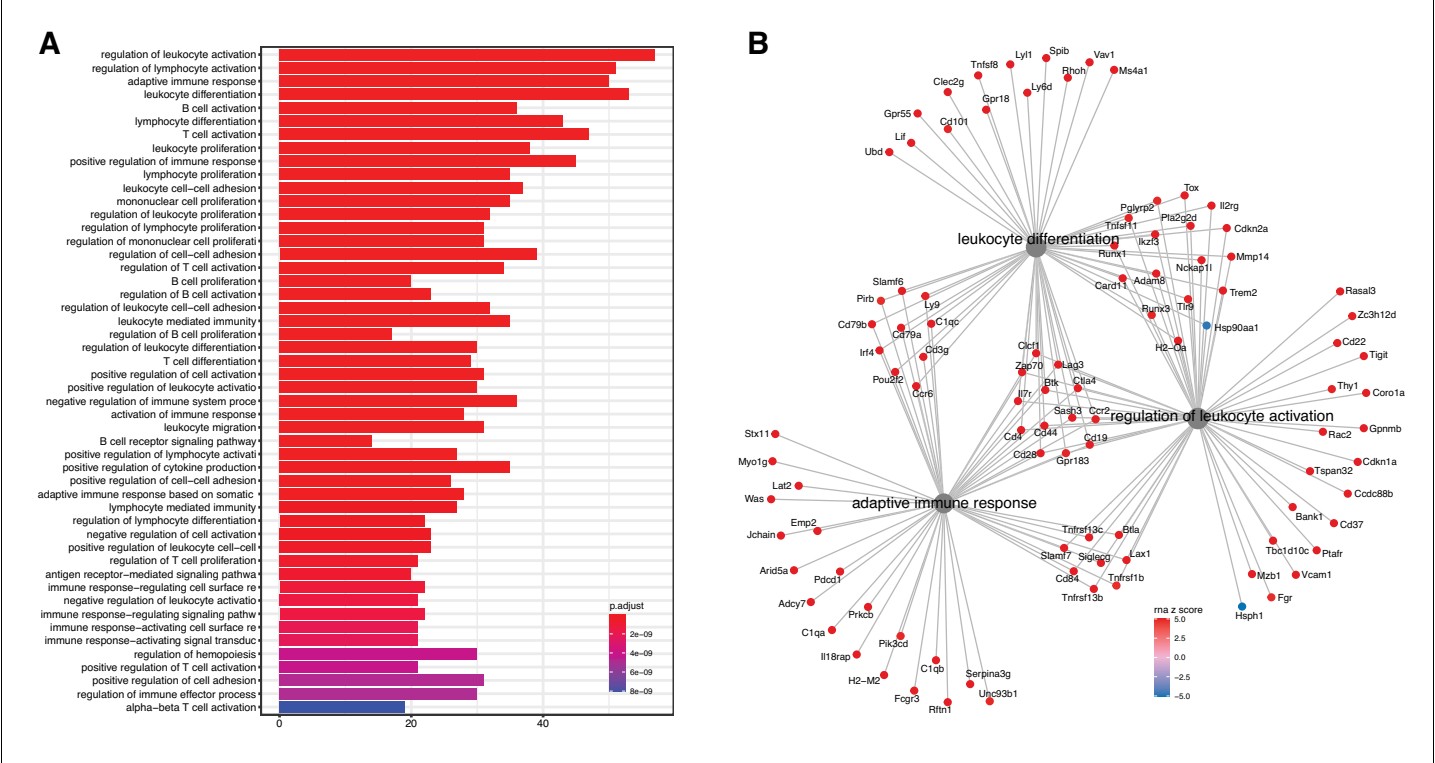

**Figure 2.** Analysis of differentially expressed mRNA with age. (**A**) Functional enrichment and (**B**) network analysis, using ClusterProfiler, show overrepresentation of genes involved in immune and inflammatory response and is likely the result of immune cell infiltration into the kidney. The online version of this article includes the following figure supplement(s) for figure 2:

**Figure supplement 1.** Relative changes in kidney cell composition as a function of age.

**Figure supplement 2.** Examples of differential mRNA expression with age.

dependent kinase genes *Cdkn1a* and *Cdkn1c*. These genes play a role in DNA damage repair and regulation of apoptosis and cellular senescence. Increased expression of other markers of cellular senescence includes *Mmp14* and *Mmp3*.

Other genes of interest with respect to renal aging and decline are *Lcn2*, encoding neutrophil gelatinase-associated lipocalin (NGAL) and *Timp1*. In patients with chronic kidney disease (CKD), NGAL closely reflects renal impairment and represents a strong and independent risk marker for progression of CKD (*Bolignano et al., 2009*). TIMP-1 has been shown to promote age-related renal fibrosis (*Zhang et al., 2006*).

Together these age-related changes in transcripts indicate infiltration of immune cells, increased cytokine activity, and cellular senescence.

## Age-related changes in protein expression

We applied a linear mixed model ANOVA to protein abundance data after applying a rank-normal scores transformation and including covariates to account for marginal effects of peptide labeling tags, generation of DO mice, and sex. We identified 876 proteins that change with age (adjusted p-value < 0.05) (*Supplementary file 2*). Of these, 352 increased with age and 524 decreased. Proteins with the most significant age trends include HIST1H1B and other histones that are decreasing with age; increasing ACTA2 a marker gene for myofibroblasts and indicator of renal fibrosis (*Duffield, 2014*); decreasing NCLN, which is part of a complex in the endoplasmic reticulum (*Haffner et al., 2007*); and decreasing CISD2, which is involved in mitochondrial autophagy (*Chen et al., 2009*; *Figure 3*).

We evaluated the proteins for functional enrichment of GO categories using ClusterProfiler and observed overrepresentation of genes associated with 186 terms (adjusted p-value < 0.01)

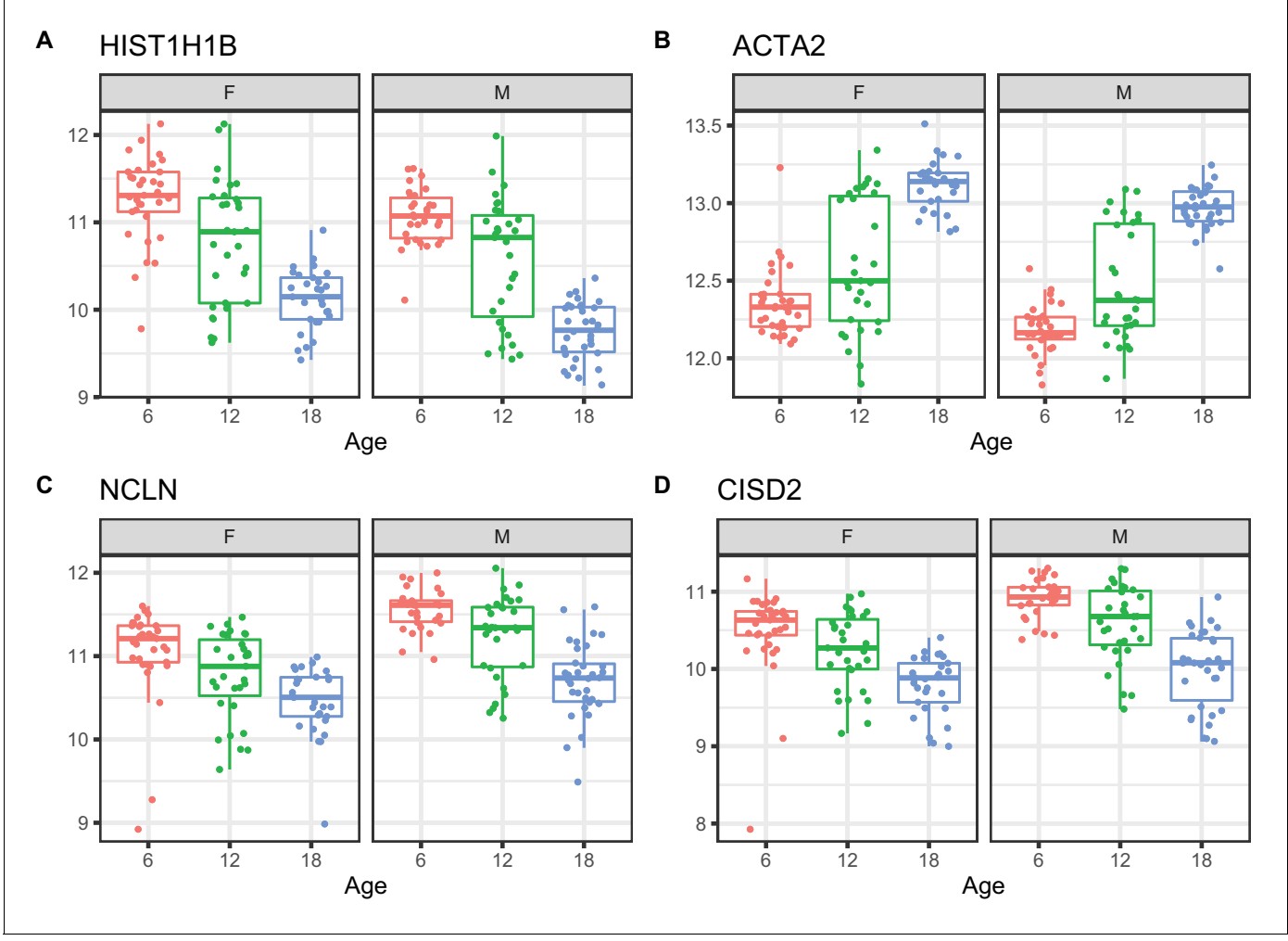

**Figure 3.** Examples of differential protein expression with age. (**A**) Decreased expression of HIST1H1B, a histone protein (**B**) increased expression of ACTA2, a marker of renal fibrosis, (**C**) decreased expression of NCLN, a part of the endoplasmic reticulum, and (**D**) decreased expression of CISD2, which is involved in mitochondrial autophagy.

The online version of this article includes the following figure supplement(s) for figure 3:

**Figure supplement 1.** Functional enrichment of differentially expressed proteins and networks.

representing a wide array of biological processes, cellular compartments, and molecular functions (*Figure 3—figure supplement 1*). We observed a general pattern of decrease with age for protein associated with oxidative phosphorylation (adjusted p-value = $1.7 \times 10^{-8}$) and the mitochondrial membrane (adjusted p-value = $2.2 \times 10^{-12}$), as well as protein exit from the endoplasmic reticulum (adjusted p-value = $3.8 \times 10^{-5}$), glycosylation (adjusted p-value = $2.6 \times 10^{-4}$), and the endoplasmic reticulum membrane (adjusted p-value = $1.1 \times 10^{-19}$). We observed an overall increase in genes associated with the actin cytoskeleton (adjusted p-value = $4.2 \times 10^{-5}$).

We observed numerous changes, both increasing and decreasing with age, in proteins associated with transmembrane transport (adjusted p-value = $1.7 \times 10^{-19}$) and the plasma membrane (adjusted p-value = $3.4 \times 10^{-6}$). These proteins are associated with mitochondrial, endoplasmic reticulum (Golgi and nuclear membranes), and the plasma membrane. Many of the latter are markers of specific cell types in the kidney. In order to better understand the mixed patterns of change, we used publicly available single-cell data sets (e.g. https://cello.shinyapps.io/kidneycellexplorer/) from mouse kidneys and literature in which proteins were localized to specific segments of the tubule using immunohistochemistry to identify genes from the enrichment categories that could be

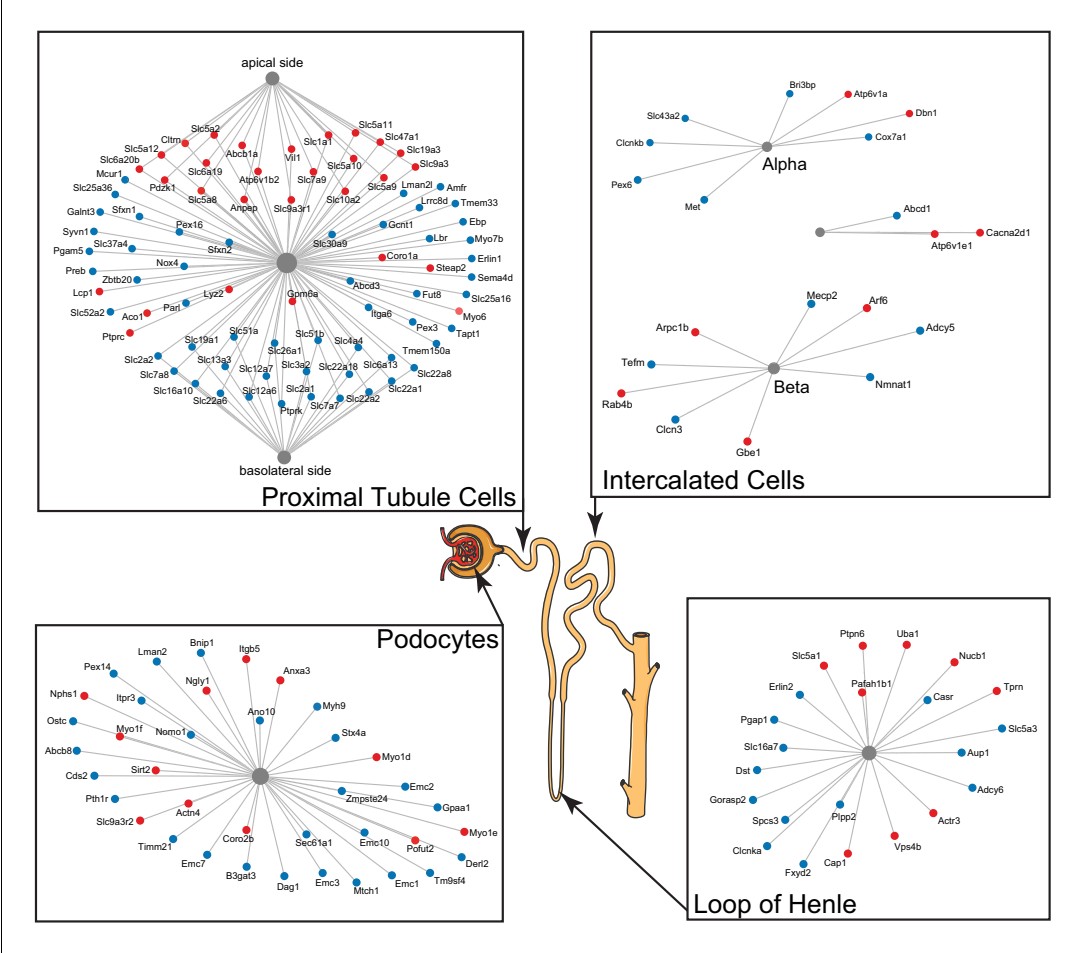

**Figure 4.** A subset of differentially expressed proteins with age that are specific for various parts of the nephron, based on single-cell data, show distinct changes with age, such as actin-cytoskeleton rearrangement in podocytes and changes in intercalated cells and the loop of Henle. Striking are the changes in channels and transporters of the proximal tubular transport, where there is an upregulation at the basolateral side and a downregulation at the apical side.

associated with specific cell types as well as the basal and apical cell surfaces of proximal tubule cells (*Figure 4*).

Two kidney-specific processes stand out: First, there is a large representation of transporters expressed in the proximal tubule cells. These are the most abundant cells in the renal tubule and do most of the work in reabsorbing salts, amino acids, glucose, and water, moving it from the filtrate back to the blood stream through transporters and channels on both the apical side and the basolateral side. We see a significant upregulation with age of the transporters on the apical side (e.g. SLC5A2 [or SGLT2], SLC7A9, SLC10A2, SLC5A8, and SLC6A19), while at the same time a significant downregulation of transporters on the basolateral side (e.g. SLC2A1 [a.k.a. GLUT1], SLC26A1, SLC4A4, SLC12A6, SLC12A7, SLC13A3, and SLC19A1). Many of these basolateral transporters depend on ion gradients generated by the Na+/K+ ATP pump for their ability to function and both subunits of this pump (ATP1A1 and ATP1B1) are significantly decreased with age. A picture of reduced proximal tubular function is completed by the overall decrease in many mitochondrial proteins, including VDAC1, which is one of the key proteins that regulate mitochondrial function and is considered the gatekeeper for the passages of metabolites, nucleotides, and ions (*Camara et al., 2017*). Transport requires a large amount of energy and proximal tubule cells are one of the most ATP-requiring cells and therefore a cell type with a very high mitochondrial content. We observe a decrease of proteins from all mitochondrial complexes as well as mitochondria-specific transporters.

The second important kidney-specific process that is suggested by the age-related changes in protein abundance is the actin remodeling in the podocytes. Remodeling of the actin cytoskeleton has a primary role in the structural adaptations made by these cells to preserve their glomerular filtration properties. Focal adhesions and slit diaphragms are signaling networks that interact with the actin cytoskeleton, maintain balance between intracellular and extracellular signals, and regulate podocyte function and morphology (*Perico et al., 2016*). For the focal adhesions, we see a decrease in dystroglycan (DAG1), a protein that is highly abundant at the interface between the podocyte foot process and the glomerular basement membrane (GBM). The slit diaphragm contains proteins that include nephrin (NPHS1) and CD2AP, both increased with age, and they regulate Cofilin (CFL1) activity, which is also increased and leads to stabilization of the actin cytoskeleton. We see an increase in actinin 4 (ACTN4), which crosslinks actin filaments in the podocyte and interacts with various other proteins. Two other upregulated proteins linking to the actin filaments and involved in filament organization are MYO1C and MYO1E. Interesting, MYH9, which also interacts with actin filaments to contract the cytoskeleton, is decreased with age. AKT2, encoding the RACβ serine/threonine-protein kinase, has an essential role in maintaining podocyte viability and normal cytoarchitecture after nephron loss (*Perico et al., 2016*). We see that AKT2 is upregulated with age. Activation of AKT2 is mediated by mTOR complex 2. Overall, these changes with age suggest actin remodeling of the podocyte foot processes, which may be due to changes in glomerular filtration rate.

Overall, we see that proteomics reveals a highly varied array of both fundamental and kidney-specific changes that are occurring at the cellular level. The picture that emerges is distinct and complementary to age-related changes observed at the transcriptional level.

## Comparison of age-related changes in mRNA and protein

In order to compare age-related trends in mRNA and proteins, we identified 6514 proteins (representing 6449 distinct genes) that had corresponding mRNA data. For purposes of this analysis, we applied a liberal false discovery rate (FDR < 0.1) multiple test correction. To provide a point of reference for interpreting the age comparisons we also compared mRNA and protein difference between the sexes.

We identified significant (FDR < 0.1) age trends in 1493 mRNAs and 3871 proteins. Direct comparison of the numbers of statistically significant mRNA and protein species that are changing with age is complicated by the potential difference in precision of the measurements as well as the different statistical testing methods. However, applying the same statistical criteria, we identified 5325 mRNAs and 4412 proteins with significant sex-specific differences. This suggests that we are not underpowered to detect changes in mRNA. We conclude that age-related changes are more prevalent for proteins as compared to mRNA, consistent with our analysis of the PCs of variation described above.

We looked at concordance in the directions of change between mRNA and protein. The absolute units of change are difficult to compare directly, so we converted the estimated fold-change (mRNA) or trend (protein) to z-scores (estimate/standard error). The correlation of age-trends between mRNA and protein (r = 0.235, p-value < $2.2 \times 10^{-16}$) is positive and highly significant but much smaller than the correlation of sex-specific differences (r = 0.626, p-value < $2.2 \times 10^{-16}$). We identified genes for mRNA and protein with age-trends that are concordant increasing (Group A, n = 309), discordant with mRNA decreasing and protein increasing (Group B, n = 196), discordant with mRNA increasing and protein decreasing (Group C, n = 108), and concordant decreasing (Group D, n = 359) (*Figure 5A*). There are more genes with concordant changes (10.3%) but a substantial proportion of genes having changes with age occur in opposite directions (4.7%). We classified sex-specific differences according to direction of change (F>M or F<M) and found 1545, 153, 610, and 1391 genes in categories A–D, respectively. Sex-specific differences are much more likely to be concordant (42.7%) with a relatively smaller proportion of discordant genes (11.7%) (*Figure 5B*).

We applied functional enrichment analysis to each group of genes defined by concordant or discordant directions of change with age (*Supplementary file 3*).

Among the 309 genes with age-related increase in both mRNA and protein (Group A), the top enrichment categories are dominated by genes involved in the immune response, actin cytoskeleton organization, and cell adhesion. As with our analysis of mRNA changes above, this likely reflects increased infiltration of immune cells into the kidney that has been previously reported in both

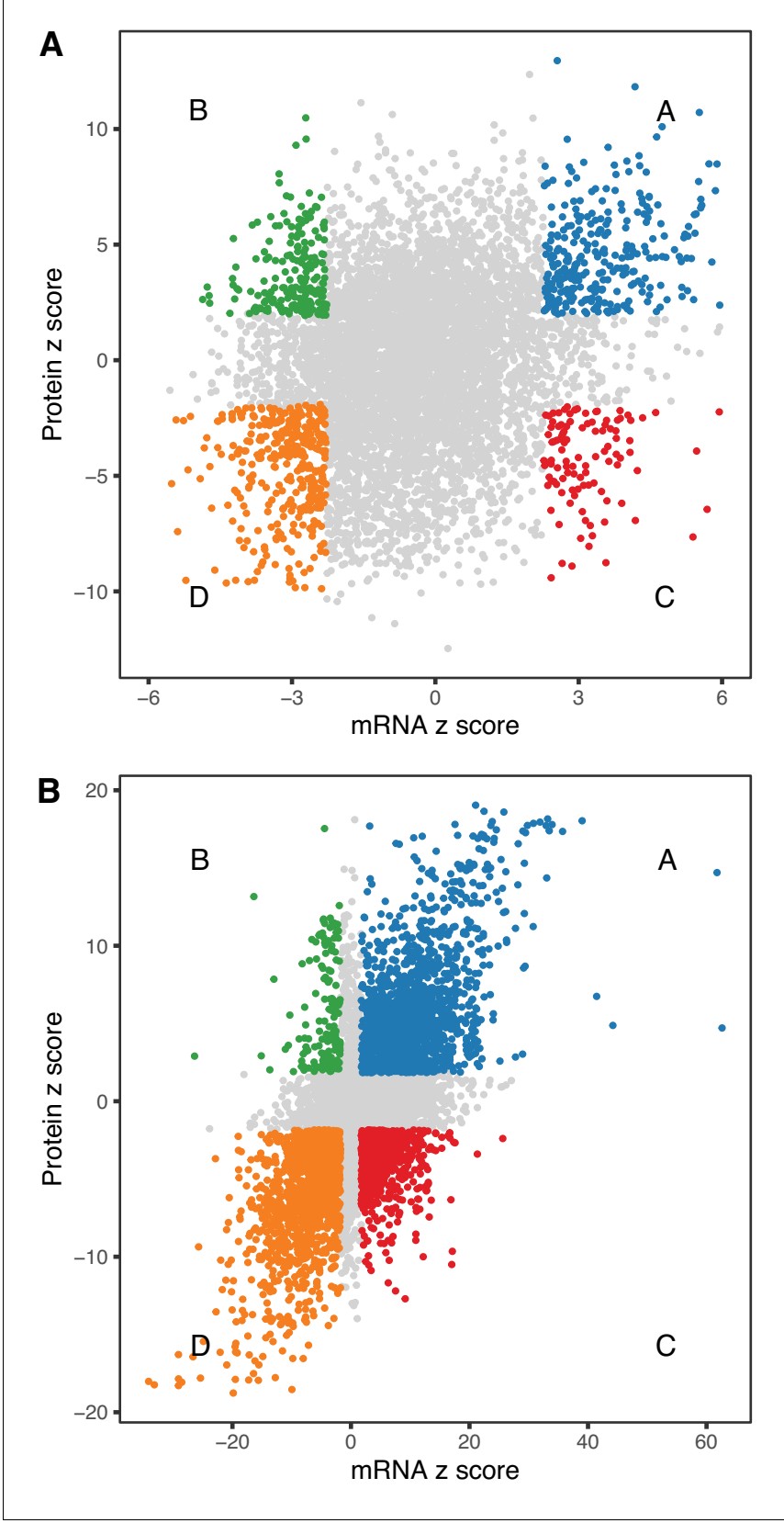

**Figure 5.** Comparison of age-related changes in mRNA and protein. We identified 6514 genes with both mRNA and protein data of which 972 had significant differences with age (**A**) for both mRNA and protein. These can be

*Figure 5 continued on next page*

*Figure 5 continued*

divided in four groups depending on the direction of change, with decreased RNA and increased protein (green), decreased RNA and decreased protein (orange), increased RNA and increased protein (blue), and increased RNA and decreased protein (red). There is a similar pattern for sex-specific differences, although these are much more likely to be concordant (B). Each point represents a gene with significantly decreased RNA and increased protein (green), decreased RNA and decreased protein (orange), increased RNA and increased protein (blue), and increased RNA and decreased protein (red).

---

humans and rodents (*Rodwell et al., 2004*; *Melk et al., 2005a*). Other genes in this group include *Slc34a2*, which encodes a sodium-dependent phosphate transporter and *Nphs1*, encoding the podocyte slit-diaphragm structural component Nephrin. We also observed concordant increases in the immuno-proteosome complex (e.g., *Psmb8*, *Psmb9*, *Psmb10*, and *Psme1*).

Among the 359 genes that had a concordant age-related decrease in mRNA and protein (Group D) we see enrichment of genes involved in transmembrane transport, mitochondrial membrane, Golgi vesicle transport that were noted previously. We also see enrichment of genes involved in junction formation including cadherins (*Cdh16*), catenins (*Ctnnb1* and *Ctnnd1*), and claudins (*Cldn8*, *Cldn10*). Cell–cell junctions are important in kidney function as much of tubular transport depends on limiting the passage of molecules and ions through the space between cells. Several adhesion-related genes such as *Aplp2* and *Ppfia1* have previously been associated with significant age-associated changes (*Rodwell et al., 2004*). Although there is no direct evidence in the kidney, it has been shown that these junctions become leaky with age in other tissues (*Jeong et al., 2017*).

Several proteins in ubiquitin mediated protein catabolism are concordantly downregulated (*Ubac2*, *Ube3c*, *Ubqln4*, *Ubr4*, *Usp20*, *Usp32*, *Usp33*, *Usp34*, *Usp39*, and *Usp9x*). Notable among the genes in this category is *Klotho*, an aging-suppressor gene primarily expressed in the kidney. *Klotho* is associated with fibrosis and is an antagonist of beta catenin (*Zhou et al., 2013*). Previous studies have shown that *Klotho* knockout mice exhibit increased incidence of aging-related diseases including atherosclerosis, vascular and tissue calcification, and CKD (*Bolignano et al., 2014*; *Kuro-o et al., 1997*). Decreased *Klotho* expression has been shown to drive accelerated aging (*Kurosu, 2005*) and renal failure (*Zhou et al., 2013*).

We observed discordant age trends between mRNA and protein for a substantial number of genes. We identified 196 genes with decreasing mRNA and increasing protein levels with age (Group B). This group includes genes that are exclusively localized to the brush border of the proximal tubule (e.g., *Slc26a6*, *Slc34a3*, *Slc5a2*, *Slc3a1*, *Slc9a3*, *Pdzk1*, and *Lrp2*) and are important for the reabsorption of specific metabolites in the ultrafiltrate. The glucose transporter SGLT2 (encoded by *Slc5a2*) is responsible for 97% of glucose reabsorption (*Vallon et al., 2011*). SLC3A1 and SLC7A9 together form the antiporter that mediates uptake of cysteine, which is of particular interest because of the potential role of cysteine in aging (*Flurkey et al., 2010*). *Lrp2* encodes Megalin, the receptor that exclusively mediates albumin reabsorption in the proximal tubule and can affect the amount of albumin in the urine (see below). PDZK1 is a scaffold protein that connects membrane proteins and regulatory components at the apical side and has been shown to interact with several transporters in Group B, such as *Slc9a3* encoding the main sodium-hydrogen antiporter (NHE3), responsible for sodium balance (*Zachos et al., 2009*), *Slc22a6*, encoding OAT1, which plays a central role in organic anion transport, and *Slc34a1* and *Slc34a3*, encoding transporters important for phosphate transport (see below) (*Lanaspa et al., 2007*).

We identified 109 genes with increasing mRNA and decreasing protein (Group C). This group is highly enriched for cytosolic ribosome including >20 genes from both small and large subunits. Group C also includes genes *Srsf3*, *Srsf5*, *Srsf6*, and *Snrpa1* that are involved in RNA splicing. Age-related changes in alternative splicing have been implicated in contributing to the dysregulation of gene expression with aging. Downregulation of splicing factor, SRSF3, has been shown to induce alternative splicing of TP53 that results in cellular senescence (*Tang et al., 2013*). It is interesting to note that for these aforementioned post-transcriptional processes, the direction of change differs between mRNA and proteins, thus hinting at possible age-related post-transcriptional regulation mechanisms at play.

Within each age category the correlations of mRNAs with their corresponding proteins are predominantly positive. Thus, the two groups of genes that show opposite directions of age-related

change between mRNA and protein appear at first to present a paradox (*Robinson, 2009*). For example, *Slc5a12*, a gene that encodes a sodium-coupled lactate transporter in the proximal tubule (Group D), shows a significant decrease in mRNA and increase in protein levels with age (*Figure 6A*). However, within each age group, we observe a positive correlation between mRNA expression and protein expression. *Flot1*, a gene in Group C, illustrates the opposite pattern of change (*Figure 6B*). Thus, while RNA and protein are usually correlated within age groups, the relative balance between RNA and protein can change in any direction across age groups. This suggests that age-related changes in protein levels are not necessarily due to transcriptional regulation and may not track age-related changes in RNA.

In order to further clarify the role of transcriptional regulation in determining age-related changes in protein, we carried out a mediation analysis between protein and corresponding mRNA for 6667 genes. We evaluated the significance of the age trend in all proteins with and without accounting for changes in mRNA (*Figure 7A*). If age-related changes in protein are fully or partially mediated by changes in mRNA, the significance of the regression of protein on age should drop after accounting for mRNA. However, we observe essentially no change in the significance of the age effect on protein and conclude that the age-related changes in protein occur independently of changes in mRNA. To provide a point of reference for our mediation analysis of the age effect on proteins, we repeated the same analysis to determine whether sex difference in protein levels could be explained by corresponding sex differences in mRNA (*Figure 6B*). Here we see that significance of the sex effect on

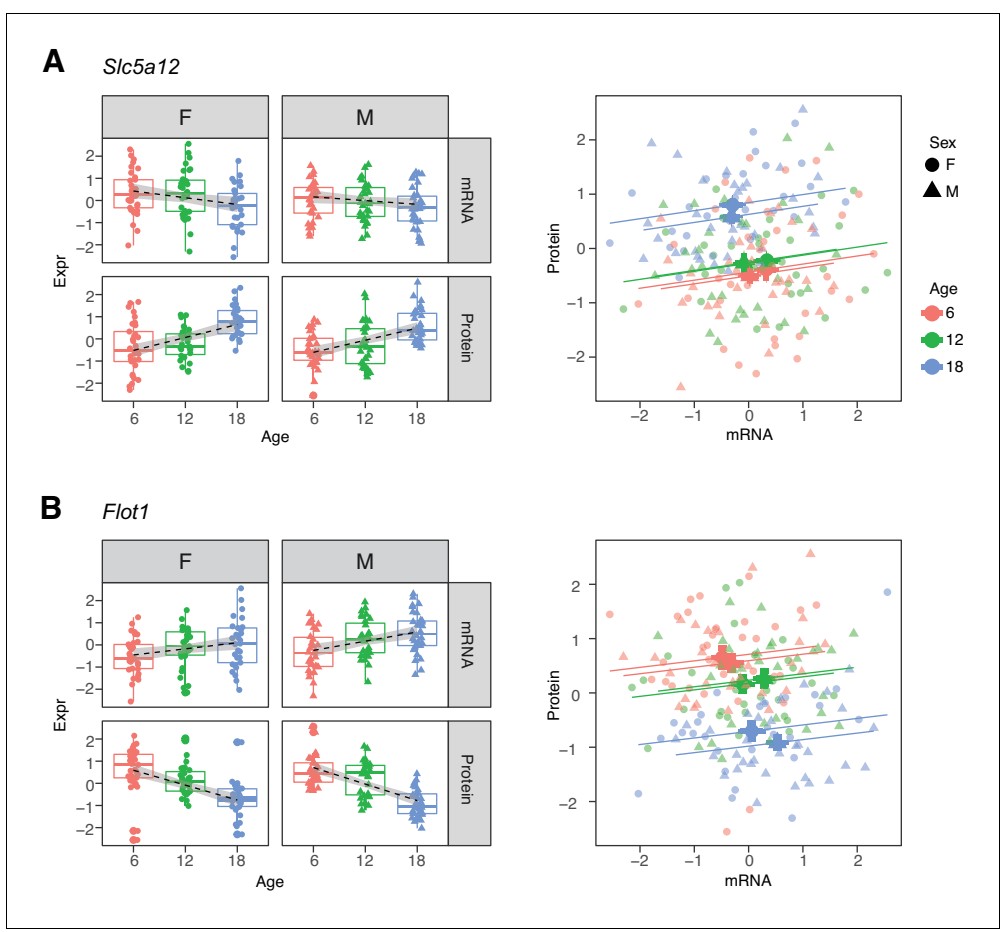

**Figure 6.** Examples of genes with opposite directions of age-related changes between mRNA and protein. (**A**) *Slc5a12* shows a decrease in mRNA and an increase in protein with age. Within age groups there is a positive correlation between mRNA and protein expression, but comparison between time points. (**B**) *Flot1* shows the opposite with increased mRNA and decreased protein with age. Females (dots) and males (triangles) at 6 (red), 12 (green), and 18 months (blue) of age.

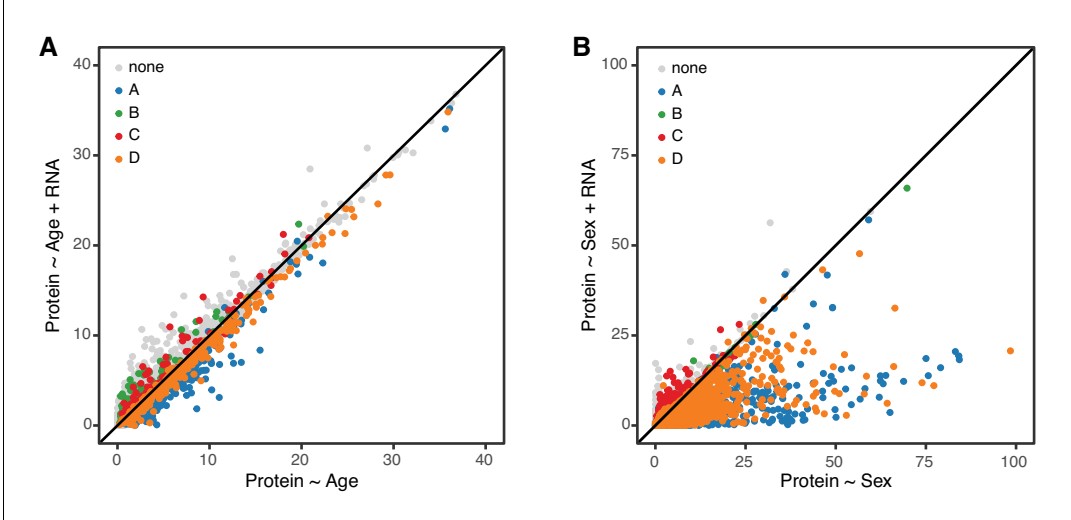

**Figure 7.** Age-related changes in protein expression are not mediated by mRNA expression. We computed the −log10 p-value from the likelihood ratio test of the age effect on protein in two regression models, without and with mRNA as a predictor. These are displayed on the x- and y-axes of the plot in panel **A**. If mRNA is a mediator of the age effect on protein, we expect to see reduced significance of the age effect when mRNA is in the regression model. Each point in the figure represents one gene (mRNA and protein) and they are colored coded to show the concordant and discordant change groups as in *Figure 4*. The points fall close to the identity line, indicating little or no drop in the −log10 p-values, and thus no evidence for mediation. Panel **B** shows a similar analysis to evaluate whether mRNA is a mediator of the effects of sex on protein. Here the −log10 p-values for most genes are reduced after including mRNA in the regression model, as expected if the effects of sex on protein are mediated through changes in mRNA.

proteins is reduced after accounting for mRNA, for most genes. This is consistent with a model in which sex-specific differences in protein are a direct response to sex-specific differences in mRNA.

For many proteins, the level of RNA expression is a major determinant of the amount of its protein product that is present. However, the age-related changes in protein amounts are not driven by a corresponding age-related change in their mRNA. What does change with age is the relative abundance of protein for a given level of mRNA expression. This suggests that the processes that drive age-related change in proteins are post-translational and may involve specific changes in translational efficiency (*Anisimova et al., 2020*) or rates of mRNA or protein turnover (*Cellerino and Ori, 2017*).

## Relating mRNA and protein changes to physiology

Regulation of urinary filtrates is a critical function of the kidney to maintain adequate concentrations in the circulating serum. Here we examine the relations of genes that were identified with significant age-related changes in both mRNA and protein levels with urinary phenotypes in order to draw further insights into the biological mechanisms underlying renal aging.

The protein most strongly correlated with urinary phosphate is *Pdzk1ip1* (r = 0.322, p-value = $7.5 \times 10^{-5}$ adjusted for sex; r = 0.32, p-value = 0.0027 adjusted for sex and age) (*Figure 8A*). *Pdzk1ip1* is expressed specifically in proximal tubule cells in segment S2 (*Lake et al., 2019*). It interacts with *Pdzk1*, which is responsible for establishing a scaffolding in the brush border of proximal tubule cells to anchor transporters and other membrane bound proteins (*Gisler et al., 2003*). *Pdzk1* and *Pdzk1ip1* are both Group B genes whose RNA decreases, and protein increases with age (*Figure 9A and B*). We fit a multivariable regression model to urinary phosphate and noted that after accounting for Pdzk1ip1 protein levels, age is no longer a significant predictor of urinary phosphate (p-value for age is 0.22 adjusted for PDZK1IP1). This is consistent with a causal for *Pdzk1ip1* as a mediator (or a close surrogate) for the change in urinary phosphate levels with age. We examined the set of 799 proteins that are significantly correlated with *Pdzk1ip1* (adjusted p-value < 0.05) and found these to be enriched for membrane transporters and other proteins

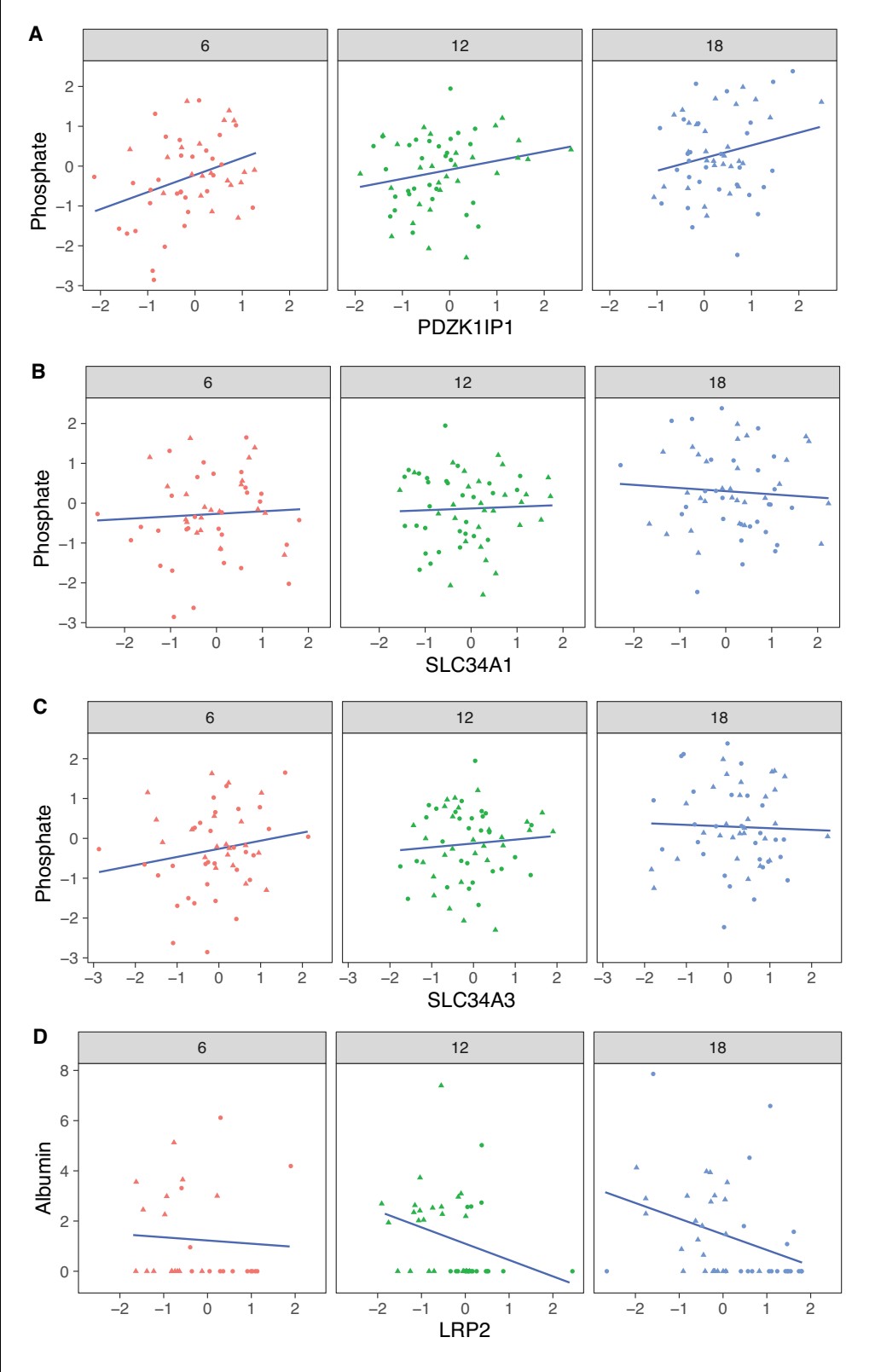

**Figure 8.** Correlation with phosphate and albumin (normalized to creatinine). (**A**) PDZK1IP1 is the protein with the highest correlation with urinary phosphate levels and mediation analysis suggests the protein is a mediator for changes in urinary phosphate levels. There are no significant correlations for phosphate with SLC34A1 (**B**) or SLC34A3 (**C**) that form the main phosphate transporter in the proximal tubule. (**D**) Significant correlation between

*Figure 8 continued on next page*

*Figure 8 continued*

albumin and its proximal tubule receptor Megalin (LRP2). Females (dots) and males (triangles) at 6 (red), 12 (green), and 18 months (blue) of age.

associated with the brush border and apical side of the cell. Interestingly, protein levels of NPT2A (*Slc34a1*) and NPT2C (*Slc34a3*) that form the major sodium-phosphate co-transporter do not correlate with urinary phosphate excretion (*Figure 8B and C*), confirming that phosphate reabsorption by this transporter is not regulated by the protein level of the subunits and instead by their interactions with PDZK1 and PDZK1IP1 (*Giral et al., 2011*).

Megalin, the protein encoded by *Lrp2*, is the receptor that mediates albumin reabsorption in the proximal tubule. It is another Group B gene that shows a decrease in both mRNA expression (p-value = $2.4 \times 10^{-5}$), an increase in protein level (p-value = 0.0043) with age and a positive correlation between mRNA and protein levels within each age group (*Figure 9C*). Megalin reabsorbs albumin that has passed through the GBM. The majority of animals in our DO cohort do not show albuminuria – 157 of the 188 animals had albumin levels below 1 mg/dL – and although this limits our ability to draw conclusions regarding the relationship between urinary albumin levels and Megalin expression levels, Megalin is negatively correlated with albumin (p-value = 0.081 after regression adjustment for sex and age) (*Figure 8D*).

## Discussion

The kidney is an excellent model for studying organ-specific aging. Its function is relatively easy to measure (through blood and urine) and functional changes can be observed early in the aging process. Although molecular changes in the aging kidney have been studied previously, these have been limited to microarray assays and RNAseq of mRNA expression. An implicit assumption of these studies is that changes in gene function (protein levels) can be extrapolated from changes in mRNA. While others have reported that mRNA can be a poor proxy for protein abundance (*Liu et al., 2016*), we find that the direction of change of a protein with age cannot be reliably predicted based on changes in its coding mRNA even for genes where the mRNA and protein are correlated within age groups. We were able to confirm previously reported transcriptional changes in the kidney with age, such as the increase in immune-related pathways due to immune cell infiltration. However, in this study we provide another layer of understanding through our data that show that the changes in the aging kidney at the molecular level are more complicated than a change in gene expression directly affecting the expression of the encoded protein in the same direction. In fact, in most cases, at the global level we show that the changes in protein expression related to aging are not mediated through mRNA levels. Through distilling this global information, we were able to obtain valuable information that distinctly show that changes in mRNA and protein can shift in both concordant and discordant direction and are pathway and cell-type dependent.

Many mechanisms known to affect the aging process can explain this disconnect between mRNA expression and protein expression. Gene-specific changes in translation rate may cause these differences. However, a recent study in which RNAseq and Riboseq data was compared in kidneys in 3-, 20-, and 32-month-old male C57BL/6 mice concluded no significant changes in translation rates between time points (*Anisimova et al., 2020*). Loss of proteostasis is a hallmark of aging that manifests at the cellular level in a number of ways, such as protein aggregation, unfolding, oxidative damage, post-translational modification, and altered rates of protein turnover (*Labbadia and Morimoto, 2014*). We recently found a decrease in correlation between protein subunits of the 26S proteasome complex in the heart of DO mice with age, suggesting that loss of stoichiometry of the proteasome system itself may contribute to the decline in the protein quality control process during aging (*Gyuricza, 2020*). Since proteostasis is tailored to the specific proteomic demands of different cells and the kidney consists of many different cell types, this is difficult to entangle in our study. Furthermore, our study can only detect protein levels and we cannot draw any conclusions about the status of the protein (functional, modified, or damaged). However, the significant changes in a number of ubiquitin-related genes and enrichment of proteins associated with the membrane of the endoplasmic reticulum with age may suggest important roles for these parts of the proteostasis network.

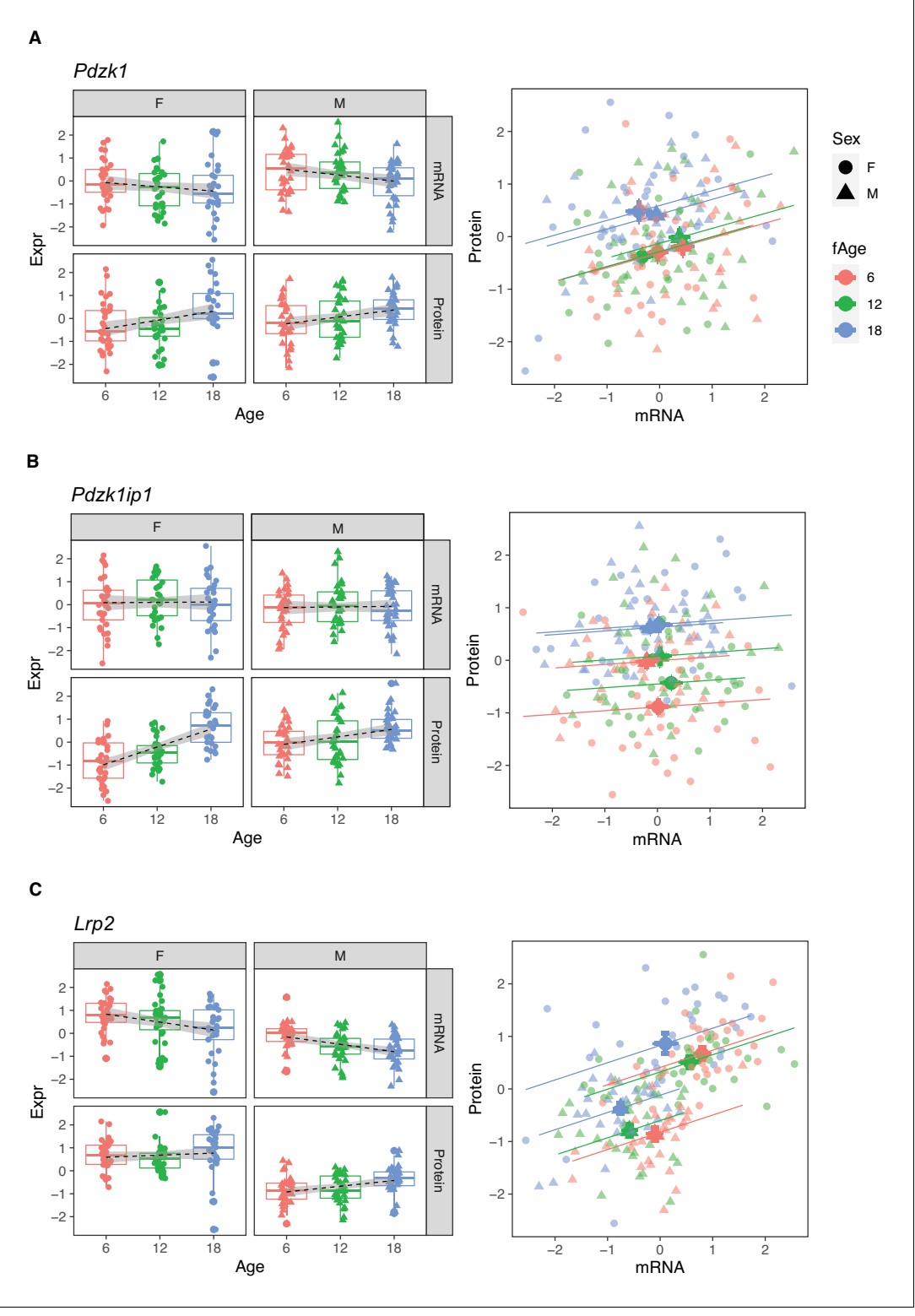

**Figure 9.** *Pdzk1* (**A**), *Pdzk1ip1* (**B**), and *Lrp2* (**C**) are genes with decreased mRNA and increased protein expression with age. Females (dots) and males (triangles) at 6 (red), 12 (green), and 18 months (blue) of age.

It is important to note that protein abundance does not equal protein function. In our study, the observation that proximal tubule-specific transporters on the apical side are increased with age, while at the same time transporters at the basolateral side are decreased poses the question what happens to cellular transport. However, increased apical-associated transporters do not necessarily mean increased uptake of molecules from the lumen and a build-up of these molecules in the cell as the transporters may not be localized to the brush border.

In conclusion, there appear to be distinct modes of aging when comparing the transcriptome and the proteome in the mouse kidney and that changes with age at the mRNA and protein level are driven by different factors. Identifying these factors and the mechanisms that disconnect gene expression from protein expression at the global scale with age will be essential to understand what drives the kidney (and possibly other organs) to become more susceptible to disease with age and to develop novel therapies to build resilience against them.

Age-related changes in protein levels are not driven by corresponding changes in mRNA. Instead they reflect a change in the balance of protein homeostasis that is altered in older animals compared to young animals. The drivers of age-specific changes do not act in the same way as factors that regulate sex differences in proteins. Sex differences appear to be largely driven by corresponding changes in mRNA. Age-related changes in protein and mRNA appear to be uncoupled, despite that protein and mRNA remain coupled within age groups. Our findings suggest that changes in protein homeostasis in aging animals occur as a result of changes in post-transcriptional processes – through changes in translation efficiency or of protein turnover. These changes appear to be coordinated across functionally related groups of genes.

# Materials and methods

## Study cohort

Our initial cohort consisted of 600 DO mice (300 males and 300 females) bred at The Jackson Laboratory (stock no. 009376) in five breeding waves (generations 8, 9, 10, 11, and 12). One hundred males and 100 females were randomly assigned to each of three groups for which tissues were collected at 6, 12, and 18 months of age. Animals were maintained on a standard rodent diet (LabDiet 5K52, St. Louis, MO, USA) in an animal room free of pathogens, at a temperature between 20°C and 22°C, and a 12 hr light:dark cycle. At the selected ages, urine samples were collected, and the right kidney was flash-frozen. Urinary albumin, phosphate, and creatinine were measured on a chemistry analyzer (Beckman Coulter AU680, Brea, CA, USA). For 188 samples, equally distributed by age and sex, flash-frozen kidney samples were pulverized and aliquoted for RNA-seq and shotgun-proteomic analysis. The mouse study was approved by The Jackson Laboratory's Institutional Animal Care and Use Committee (AUS#06005).

## DNA isolation and whole-genome diplotype probability construction

Tail tips were collected and DNA was isolated using standard methods. DNA concentration and purity were measured using a NanoDrop 2000 spectrophotometer (Thermo Fisher Scientific, Waltham, MA, USA). Samples met stringent quality standards of A260/280 ratio between 1.7 and 2.1. Mice were fully genotyped for 78,000 SNPs using the GeneSeek Mega Mouse Universal Genotyping Array (MegaMUGA) (Neogen Genomics, Lincon, NE, USA) (*Morgan et al., 2016*). Founder haplotype mosaics were reconstructed using a Hidden Markov Model of array intensity data generated from the BeadStudio (Illumina, San Diego, CA, USA) algorithm. To ensure quality of genotype construction, samples with call rates of 90% and over were kept.

## Total RNA isolation and quality control

The pulverized whole kidney samples were lysed and homogenized in Ambion TRIzol reagent (Thermo Fisher Scientific #15596026). Total RNA was isolated using miRNeasy Mini kit (Qiagen Inc #217004, Germantown, MD, USA) according to manufacturer's protocols, including the optional DNase digest step. Sample concentration and quality were assessed using the Nanodrop 2000 spectrophotometer and RNA 600 Nano LabChip assay (Aligent Technologies, Santa Clara, CA, USA).

## Library construction and high-throughput RNA sequencing

Poly(A) RNA-seq libraries were constructed using the TruSeq RNA Library Prep Kit v2 (Illumina), including the addition of unique barcode sequences. Library quality and quantity were assessed using the DNA 1000 LabChip assay (Agilent Technologies) and quantitative PCR (Kapa Biosystems, Wilmington, MA, USA). Eight pools of 24 randomized libraries were sequenced in three lanes at 100 bp single-end on the HiSeq 2500 (Illumina) using TruSeq SBS Kit v4 reagents (Illumina).

## High-throughput proteomics

Samples of the same mice that were used for RNA-seq were homogenized in 1 ml lysis buffer, which consists of 1% SDS, 50 mM Tris, pH 8.8 and Roche complete protease inhibitor cocktail (Roche # 11697498001, Clifton, NJ, USA) and analyzed as previously described (*Chick et al., 2016*).

## Quantification and testing of RNA-seq data

Genotyping by RNA-Seq software (https://gbrs.readthedocs.io/en/latest/) was used to align the RNA-Seq reads and reconstruct the individual haplotypes of DO mice. GBRS-constructed haplotypes were cross-compared against MegaMUGA-constructed diplotypes as a confirmation step to identify and correct sample mix-ups (*Broman et al., 2019*). We applied Expectation-Maximization algorithm for Allele Specific Expression (EMASE) (*Raghupathy et al., 2018*) to quantify gene expression from the individual aligned RNA-seq data. Count data were normalized using DESeq2 (*Love et al., 2014*) variance stabilizing transformation. Differential expression testing was done with DESeq2 using a likelihood ratio test to evaluate changes with age as a linear trend. The trend test compares two linear predictors, one with terms for sex and sequencing batch and a second with an additional term for age. Age is coded using centered and scaled values (−0.5, 0, 0.5) to obtain effect size estimates in units of $\log_2$ fold change per year. We chose to test for a linear trend because the test is more powerful than the general test for age as a three-level categorical variable. The trend test will detect nonlinear changes that have an overall trend, but with only three time points in the data, modeling nonlinear effects is not feasible.

## Quantification and testing of protein expression data

Tissue from the total (right) kidney samples was homogenized in 1 ml lysis buffer, which consisted of 1% SDS, 50 mM Tris, pH 8.8 and Roche complete protease inhibitor cocktail (Roche # 11697498001, Clifton, NJ, USA), and analyzed as previously described (*Chick et al., 2016*). Protein abundances were estimated from their component peptides identified through mass spectrometry (MS) followed by a database search of MS spectra. Prior to protein expression estimation, we filtered out peptides that contained polymorphisms relative to the mouse reference genome. We determined the age-related changes for each protein by computing the likelihood ration test for age (as a linear trend) using a mixed model linear regression with random effect terms to account for protein labeling (Tag), DO mouse generation, and a fixed-effect correction for sex. The regression coefficient and p-values generated from this model were used to determine the significant change in direction of the mRNA and proteins.

For both mRNA and protein data, we applied two types of multiple test corrections. We applied a stringent family-wise error correction using Holm's method (https://www.jstor.org/stable/4615733), indicated as 'adjusted p' in the text. We applied a less stringent false discovery rate correction using the Benjamini–Hochberg method (http://www.jstor.org/stable/2346101), indicated as 'FDR' in the text.

## In silico cell type deconvolution of bulk RNA-Seq

In order to examine relative changes in cell composition with age, we used in silico cell type deconvolution as implemented in the CellCODE R package (*Chikina et al., 2015*). This approach uses marker genes derived from transcriptional profiles ascertained for purified cell types to quantify changes in cell composition in a heterogeneous mixture of RNA (bulk RNA-Seq). For this analysis we used a bulk gene expression matrix that was upper quartile normalized to account for differences in library size. We obtained cell type-specific transcriptional profiles using single cell RNA-Seq data and cell type labels from *Park et al., 2018*. To obtain markers from single cell transcriptional profiles, we normalized the count matrix by the number of unique molecular identifiers (UMI) per cell

per 10,000 UMIs, took the log of counts + 1, and calculated mean expression of each gene in each cell type. We allowed up to 50 markers per cell type and calculated surrogate proportion variables using the 'raw' method to ensure that trends in relative cell type proportions were estimated independently of mouse age (*Chikina et al., 2015*).

## Mediation analysis

We evaluated the potential for mRNA to act as a mediator of age-related changes in protein. We applied a regression modeling approach to the mediation analysis. We evaluated the significance of the regression of protein on age with and without mRNA as a predictor. If mRNA is a mediator of the effects of age on protein, we expect a substantial drop in the significance of the age term when mRNA is included in the model. There are alternative explanations for a drop in significance, but if mRNA fully or partially mediates the age effect, we should see the drop. To see the mediation effect, we compared $-\log_{10}$ p-values for the age term in linear regression models Protein ~ Age + Sex + Generation and Protein ~ mRNA + Age + Sex + Generation. To evaluate whether mRNA could be mediating the effect of sex on proteins, we repeated the mediation analysis but computed significance of the sex term in the linear regressions.

## Data normalization and statistics

All RNA-seq expression data and proteomics data were transformed to rank-normal scores (*Conover, 1999*) prior to analysis, unless otherwise stated. Albumin and urinary phosphorus data were log transformed after regression adjustment for urinary creatinine levels. Data analysis and figures were generated using R v4.0.0.

## Sample size determination and allocation of samples to treatments

The sample size required to detect a significant age trend is determined by the expected size of the trend (difference in means between the 6- and 18-month age groups) relative to the within age-group variance. Therefore, we define the strength of a trend in units of standard deviation (SD) of the within group variance. Based on standard power calculations (*Voorhis and Morgan, 2007*), with a sample size of ~64 animals per age group we can expect to achieve power = 0.80 to detect an age trend of 0.5 SD at an unadjusted type I error of 0.05. In practice because of the high precision of the RNA and protein quantification, this enabled us to detect age trends in the majority of genes tested. In order to focus on only the most substantial trends in the data we applied a strong, family-wise error rate correction for multiple testing (https://www.jstor.org/stable/4615733).

The experiment was performed once. Both RNA and protein measurements were carried out in batches that were accounted for in the analysis. Samples were assigned to batches at one time in fully randomized order. Sample labels were not masked. There were no technical replications performed. In order to minimize the impact of outliers that inevitably occur in large-scale data, we analyzed after rank normal scores transformation (*Conover, 1999*). We included only samples for which we had obtained both RNA and protein data.

Mice were assigned to pens at random with four mice per pen at weaning. The pens were assigned to age groups and tissue collection dates at random at weaning. The full study was populated across six generations of DO breeding spanning a period of 2 years. Assignment of pens to age groups was partially balanced across generations to minimize confounding effects and to ensure that tissue collection dates were spread evenly across the calendar year.

## Acknowledgements

We gratefully acknowledge the contribution of Heidi Munger and the Genome Technologies Service at The Jackson Laboratory for expert assistance with the work described in this publication. This work was supported by the National Institutes of Health grant to The Jackson Laboratory Nathan Shock Center of Excellence in the Basic Biology of Aging (AG038070).

## Additional information

### Funding

| Funder | Grant reference number | Author |
| --- | --- | --- |
| National Institutes of Health | AG038070 | Ron Korstanje |
| National Institutes of Health | AG038070 | Gary A Churchill |

The funders had no role in study design, data collection and interpretation, or the decision to submit the work for publication.

### Author contributions

Yuka Takemon, Data curation, Formal analysis, Writing - original draft, Writing - review and editing; Joel M Chick, Investigation, Methodology; Isabela Gerdes Gyuricza, Methodology, Writing - review and editing; Daniel A Skelly, Formal analysis, Methodology; Olivier Devuyst, Methodology; Steven P Gygi, Conceptualization, Funding acquisition, Investigation, Methodology; Gary A Churchill, Conceptualization, Formal analysis, Funding acquisition, Investigation, Methodology, Writing - original draft, Writing - review and editing; Ron Korstanje, Conceptualization, Supervision, Writing - original draft, Project administration, Writing - review and editing

### Author ORCIDs

Yuka Takemon (iD) http://orcid.org/0000-0002-3538-4409
Isabela Gerdes Gyuricza (iD) http://orcid.org/0000-0002-7969-1910
Daniel A Skelly (iD) http://orcid.org/0000-0002-2329-2216
Olivier Devuyst (iD) http://orcid.org/0000-0003-3744-4767
Gary A Churchill (iD) http://orcid.org/0000-0001-9190-9284
Ron Korstanje (iD) https://orcid.org/0000-0002-2808-1610

### Ethics

Animal experimentation: The Jackson Laboratory's Institutional Animal Care and Use Committee approved all reported mouse studies (AUS#06005).

### Decision letter and Author response

Decision letter https://doi.org/10.7554/eLife.62585.sa1
Author response https://doi.org/10.7554/eLife.62585.sa2

## Additional files

### Supplementary files

• Supplementary file 1. List of genes with significant age-related, sex-related, and age-by-sex interaction-related changes in mRNA expression levels.

• Supplementary file 2. List of genes with significant age-related, sex-related, and age-by-sex interaction-related changes in protein expression levels.

• Supplementary file 3. List of genes with significant age-associated changes in both mRNA and protein levels and their direction of change.

• Transparent reporting form

### Data availability

Source data and analysis scripts have been deposited with FigShare (https://doi.org/10.6084/m9.figshare.12894146.v1). In addition, the transcript and protein data are available in an online tool that supports genetic mapping analysis (https://churchilllab.jax.org/qtlviewer/JAC/DOKidney). The RNA-seq data have been deposited in NCBI's Gene Expression Omnibus, accession number GSE121330 (https://www.ncbi.nlm.nih.gov/geo/query/acc.cgi?acc=GSE121330). The mass spectrometry

proteomics data have been deposited to the ProteomeXchange Consortium via the PRIDE partner repository with the dataset identifier PXD023823.

The following datasets were generated:

| Author(s) | Year | Dataset title | Dataset URL | Database and Identifier |
|---|---|---|---|---|
| Takemon Y, Churchill G, Korstanje R | 2019 | Transcriptomic profiling reveals distinct modes of aging in the kidney | https://www.ncbi.nlm.nih.gov/geo/query/acc.cgi?acc=GSE121330 | NCBI Gene Expression Omnibus, GSE121330 |
| Churchill GA | 2020 | Source Data for Aging Kidney Project | https://churchilllab.jax.org/qtlviewer/JAC/DO-Kidney | Aging Kidney Project, DOKidney |
| Churchill GA | 2020 | Source Data for Aging Kidney Project | https://doi.org/10.6084/m9.figshare.12894146.v1 | figshare, 10.6084/m9.figshare.12894146.v1 |
| Churchill GA | 2021 | Proteomic and transcriptomic profiling reveal different aspects of aging in the kidney | https://www.ebi.ac.uk/pride/archive/projects/PXD023823 | PRIDE, PXD023823 |

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
