## [Decision Letter]

**Acceptance summary:**

The co-profiling of proteomic and transcriptomic changes of kidney aging in genetically diverse mice at different ages generated a rich resource for aging research. The common and unique mRNA and protein changes associated with kidney aging provides a more comprehensive picture of the aging processes.

**Decision letter after peer review:**

Thank you for submitting your article "Proteomic and transcriptomic profiling reveal different aspects of aging in the kidney" for consideration by *eLife*. Your article has been reviewed by two peer reviewers, and the evaluation has been overseen by a Reviewing Editor and Jessica Tyler as the Senior Editor. The following individual involved in review of your submission has agreed to reveal their identity: Steve Horvath (Reviewer #2).

Summary:

The reviewers have discussed the reviews with one another and the Reviewing Editor has drafted this decision to help you prepare a revised submission. In this work, the authors measured kidney mRNA and protein levels in 188 genetically diverse mice at ages 6, 12, and 18 months. They found age-related changes in both mRNA and protein were associated with increased immune infiltration and decreases in mitochondrial function. In addition, they observed some age-related changes in protein showed no corresponding changes in mRNA. Therefore, the authors concluded that examination of changes in proteins is essential to understand aging processes that are not transcriptionally regulated. Overall the experiments are well designed and provides the research community a new dataset of both gene expression and protein expression data to study murine kidney aging.

Essential revisions:

1) The figure legends were often too brief and did not provide sufficient information to help readers to understand the figure. For example, in Figure 6, it was not clear each dot's meaning, and the figure legend was not very helpful in general.

2) Some of the analyses were not well described. For example, DEseq2 is most commonly used to compare differential gene expressions under two conditions; however, in this experiment, there were three time points. Additional description will be helpful to demonstrate how the trend test was performed and what exactly was tested. It seems that there were way many more up-regulated genes than down-regulated genes from this analysis. However, using a different model, when authors compared gene expression with protein expression, there were comparable numbers of up-regulated genes vs. down-regulated genes. Would non-linear changes be captured by this analysis, e.g., up- regulation in the second time point then down-regulation in the third time point?

3) In the subsection “Age-related changes in Protein Expression”, the authors tried to identify proteins/genes from enrichment categories that were associated with specific cell types as illustrated in Figure 3. However, it is not clear how accurate such cell type inference was. Additional information such as how cell type specific marker genes were derived will be helpful to understand what the authors did to decompose the proteomic changes into cell-type specific changes as suggested in the aforementioned subsection and Figure 3.

4) The authors observed essentially no change in the significance of correlations between protein and age when mRNA expression was considered or excluded from the regression model. Therefore they concluded that the age-related components of change in protein abundance occurred independently of the mRNA. This is not fully anticipated, as I would expect that for some proteins, their age-related changes depend on mRNA. It seems to me that a conditional independence test would be more appropriate, i.e., to test if protein and age are still correlated when condition on mRNA expression.

---

## [Author Response]

Essential revisions:1) The figure legends were often too brief and did not provide sufficient information to help readers to understand the figure. For example, in Figure 6, it was not clear each dot's meaning, and the figure legend was not very helpful in general.

We have rewritten and expanded the figure legends.

2) Some of the analyses were not well described. For example, DEseq2 is most commonly used to compare differential gene expressions under two conditions; however, in this experiment, there were three time points. Additional description will be helpful to demonstrate how the trend test was performed and what exactly was tested. It seems that there were way many more up-regulated genes than down-regulated genes from this analysis. However, using a different model, when authors compared gene expression with protein expression, there were comparable numbers of up-regulated genes vs. down-regulated genes. Would non-linear changes be captured by this analysis, e.g., up- regulation in the second time point then down-regulation in the third time point?

DESeq is a powerful tool that can do much more than two sample comparisons – for example, it can compute a likelihood ratio test comparing two nested models. In this case we provided two model formulas, one with just covariates and the other with covariates plus Age expressed as a continuous variate (not a factor), centered and in units of years (-0.5, 0, 0.5). The likelihood ratio test for these models will be significant if there is a trend (up or down) with Age. DESeq2 will also estimate the magnitude of the trend in units of log_2_ fold change per year. We have added text to the Materials and methods to clarify how this was done.

At the stringent significance levels (Holm adjusted p < 0.05), there are 449 RNAs that have a significant Age effect. This is a very stringent criteria that picks out only those RNAs with the biggest trends. Of these, 426 are increasing (95%!). However, if we relax the stringency using a false discovery rate (FDR < 0.1), as we will do in later sections of the manuscript, we find 4039 RNAs are changing with Age. Of these, 2649 are increasing (65%). We would interpret this to mean that there is a slight bias for RNAs that change with Age to be increasing – and the bias gets stronger as the significance becomes more stringent. We checked using magnitude of fold-change and the same pattern holds, RNAs with the biggest change tend to be increasing with Age, it is such a striking observation, we have added text to explain further (Results).

It is likely that many of the Age-related changes in RNA (and protein) are non-linear. But we only have data at three time points. The trend test (with 1 df, Age as a variate) is more powerful than the general test (2df, Age as a factor). The trend test will pick up trends, even if the change is non-linear – unless they go up and down by the same magnitude (or down then up). Non-linear changes would be of interest, but with only three time points we have little confidence in our ability to detect and characterize them. See the Materials and methods.

3) In the subsection “Age-related changes in Protein Expression”, the authors tried to identify proteins/genes from enrichment categories that were associated with specific cell types as illustrated in Figure 3. However, it is not clear how accurate such cell type inference was. Additional information such as how cell type specific marker genes were derived will be helpful to understand what the authors did to decompose the proteomic changes into cell-type specific changes as suggested in the aforementioned subsection and Figure 3.

We used published single-cell datasets from several studies and literature in which these genes were localized using immunohistochemistry. We have added sentences in the Results and in the legend of Figure 3.

4) The authors observed essentially no change in the significance of correlations between protein and age when mRNA expression was considered or excluded from the regression model. Therefore they concluded that the age-related components of change in protein abundance occurred independently of the mRNA. This is not fully anticipated, as I would expect that for some proteins, their age-related changes depend on mRNA. It seems to me that a conditional independence test would be more appropriate, i.e., to test if protein and age are still correlated when condition on mRNA expression.

We agree that this is a surprising finding, and that is why we set the stage by looking in detail at how RNA-protein pairs change with age. We conclude that, even for genes with discordant directions of change, RNA and protein are usually correlated within age groups. However, the average level of RNA and average level of protein can change across age groups in any direction. This is best illustrated in Figure 5. RNA and protein can still be coupled but display different directions of change with age. We have added text to expand on this point (Results).

The regression test – comparing models for protein with vs. without RNA as a predictor – is the likelihood ratio test for conditional independence. The way we present it is not standard. On the x- and y-axes of the plot in Figure 6 we show the -log10 p-value for the effect of Age (on protein) without and with RNA in the model, respectively. If RNA is a complete mediator, i.e., Age -> RNA -> Protein, the adjusted -log 10 p-values (y-axis) would drop to nearly 0, i.e., we fail to reject the null hypothesis of conditional independence. The conditional independence model is a null hypothesis so we cannot “prove” it. The situation is trickier because if there is any measurement error (there always is), complete mediation doesn’t hold – but the -log10 p-values should still drop to the extent that (measured) RNA is a partial mediator. Complicated. As a sanity check, we also evaluated whether RNA could be a mediator of the effects of Sex on protein, Sex -> RNA -> Protein. Sex effects appear to be at least partially mediated by RNA for most genes. We checked everything three times, there is no mistake. We have expanded our explanation in the text (Materials and methods) and also expanded the legend of Figure 6 to (partially) explain the logic of this analysis. We feel that this is potentially an important finding with implications about the nature of the aging process – but requires more follow-up with data spanning different species and contexts.